# Molecular epidemiology and antimicrobial resistance phenotype of paediatric bloodstream infections caused by Gram-negative bacteria

Sam Lipworth [1,2✉], Karina-Doris Vihta [1,3], Tim Davies[1,2], Sarah Wright[2], Merline Tabirao[2], Kevin Chau[1], Alison Vaughan[1], James Kavanagh [1], Leanne Barker[1], Sophie George[1], Shelley Segal[2], Stephane Paulus[2], Lucinda Barrett[2], Sarah Oakley[2], Katie Jeffery [2], Lisa Butcher[2], Tim Peto[1,2,4,5], Derrick Crook[1,2,4,5], Sarah Walker[1,4,5], Seilesh Kadambari[2,6,7] & Nicole Stoesser [1,2,7✉]

## Abstract

**Background** Gram-negative organisms are common causes of bloodstream infection (BSI) during the neonatal period and early childhood. Whilst several large studies have characterised these isolates in adults, equivalent data (particularly incorporating whole genome sequencing) is lacking in the paediatric population.

**Methods** We perform an epidemiological and sequencing based analysis of Gram-negative bloodstream infections (327 isolates (296 successfully sequenced) from 287 patients) in children <18 years old between 2008 and 2018 in Oxfordshire, UK.

**Results** Here we show that the burden of infection lies predominantly in neonates and that most infections are caused by *Escherichia coli, Klebsiella* spp. and *Enterobacter hormaechei*. There is no evidence in our setting that the proportion of antimicrobial resistant isolates is increasing in the paediatric population although we identify some evidence of sub-breakpoint increases in gentamicin resistance. The population structure of *E. coli* BSI isolates in neonates and children mirrors that in adults with a predominance of STs 131/95/73/69 and the same proportions of O-antigen serotypes. In most cases in our setting there is no evidence of transmission/point-source acquisition and we demonstrate the utility of whole genome sequencing to refute a previously suspected outbreak.

**Conclusions** Our findings support continued use of current empirical treatment guidelines and suggest that O-antigen targeted vaccines may have a role in reducing the incidence of neonatal sepsis.

## Plain language summary

Bacterial bloodstream infections cause serious disease in newborns and older children. However, compared to adults, there have been comparatively few studies in children that look at the genetic sequence of the bacteria causing these infections and the antibiotic resistance genes that they carry. In this study, we analysed the genetic sequences of bacteria from bloodstream infections occurring over a 10-year period in children in Oxfordshire, UK. We found that, unlike in adults, infections caused by the two most common bacteria do not seem to be occurring more frequently and that rates of antibiotic resistance associated with them are also stable. We also found that vaccines currently in development to protect against severe *E. coli* disease in adults should, at least in theory, also work in children and so further research is needed to further explore this possibility.

[1] Nuffield Department of Medicine, University of Oxford, Oxford, UK. [2] Oxford University Hospitals NHS Foundation Trust, Oxford, UK. [3] Department of Engineering, University of Oxford, Oxford, UK. [4] NIHR Health Protection Research Unit in Healthcare Associated Infections and Antimicrobial Resistance at University of Oxford in partnership with Public Health England, Oxford, United Kingdom. [5] NIHR Oxford Biomedical Research Centre, Oxford, United Kingdom. [6] Department of Paediatrics, University of Oxford, Oxford, UK. [7] These authors jointly supervised this work: Seilesh Kadambari, Nicole Stoesser. ✉email: Samuel.lipworth@ndm.ox.ac.uk; Nicole.stoesser@ndm.ox.ac.uk

Gram-negative bloodstream infections (GNBSI) are a common cause of substantial morbidity and mortality globally in neonates and young children[1–4]. Their incidence has increased in both the UK and the US over the past decade, particularly in very low birth-weight neonates (VLBW, defined as <1500 g)[5,6]. Their association with antimicrobial resistance (AMR) has been highlighted by a recent study in the United States of 721 *E. coli* isolates (including 393 bloodstream infection [BSI]-associated isolates), which found high rates of non-susceptibility to commonly used empirical antibiotics, including ampicillin (66.8%) and gentamicin (16.8%), as well as an extended beta-lactamase (ESBL) phenotype in 1 in 20 cases[7]. A recent study of 2483 neonates with culture-confirmed sepsis in low and middle-income countries showed that *Klebsiella* spp. was the predominant pathogen causing multidrug-resistant neonatal sepsis[8]. In Greece, a retrospective observational study in 16 neonatal intensive care units (NICUs) revealed almost half (45%; 36/80) of *Klebsiella* spp. were resistant to either gentamicin or amikacin[9]. The ability of many Gram-negative bacilli (GNB) to readily acquire and exchange genetic material (particularly AMR genes [ARGs]) via mobile genetic elements means that the proliferation of drug-resistant strains remains a constant threat.

The molecular epidemiology of *E. coli* and *Klebsiella* spp. isolates causing invasive infection in adults has been characterised in large sequencing studies.[10,11] These have demonstrated the emergence of particular AMR-associated sequence types (e.g., *E. coli* ST131)[12], the genetic homogeneity of isolates causing community and nosocomial onset infections suggesting a common reservoir[13], the potential for vaccines to play a role in reducing the incidence of these infections[14,15], and the emerging threat of the convergence of multidrug resistance and hypervirulence in *Klebsiella* spp[16]. To our knowledge, no study to date has systematically evaluated the molecular epidemiology of *E. coli*/*Klebsiella* spp. and other common causes of GNBSI in a paediatric population; published studies focus predominantly on evaluations of outbreaks caused by AMR-associated strains and/or on neonates (see above)[8,17,18]. In this study, we, therefore, aimed to investigate sequencing data from a relatively large collection of sequentially acquired, unselected bloodstream isolates from neonates and children presenting to hospitals in Oxfordshire, UK, over the past decade. We find that GNBSIs in this population are primarily caused by *E. coli*/*Klebsiella* spp./*Enterobacter hormaechei*, with the greatest burden of disease occurring in neonates; overall, there were no substantial changes in antimicrobial susceptibility, supporting the continued use of current empirical regimens. We demonstrate that the population structure of *E. coli* in the paediatric population mirrors that seen in adults and demonstrate the utility of WGS to monitor determinants of AMR and the genomic relatedness of isolates, leading us to refute a previously suspected outbreak.

## Methods

**Isolate selection.** Oxford University Hospitals NHS Foundation Trust is a large healthcare provider in the South East of England, serving a paediatric population of ~142,000 across four hospitals (of which two have emergency and acute general paediatric medicine, and one provides all neonatal/paediatric critical care and specialist paediatric services for the region). The microbiology laboratory additionally provides a service to all regional community healthcare providers. All *E. coli* and *Klebsiella* spp. isolates (deduplicated to one morphotype per 90-day period) from Oct-2008 to Nov-2018 collected from blood cultures of patients <18 years old on the day of collection were included in the study. The same selection criteria were applied to other GNB from August 2011 to September 2018, which were excluded from the initial period due to resource limitations. Prior to 2013, antimicrobial susceptibility testing was performed using disk diffusion; after this, the Phoenix BD system was used with European Committee on Antimicrobial Susceptibility Testing (EUCAST) breakpoints. Amikacin susceptibility phenotyping was not routinely performed prior to 2013.

**Sequencing procedures.** Frozen stocks were sub-cultured onto Columbia blood agar and incubated overnight at 37 °C. DNA extractions were performed using the QuickGene DNA extraction kit (Autogen, MA, USA) as per the manufacturer's instructions (with an additional mechanical lysis step—FastPrep, MP Biomedicals, CA, USA; 6 m/s for 40 s, done twice). Sequencing was performed using Illumina HiSeq 2500/3000/4000/MiSeq instruments as described previously[19]. Briefly, library preparation was performed using the NEBNext Ultra DNA Sample Prep Master Mix Kit (New England Biosciences). Illumina Multiplex Adaptors were used to create ligated libraries with subsequent size selection using Agencourt Ampure magnetic beads (Beckman Coulter), followed by PCR-based enrichment and adaptor extension. We purified the products using Agencourt Ampure XP beads (Beckman Coulter) using a Biomek NXp per the manufacturers' instructions. We used a Tapestation system to assess size distributions of library preparations and quantified their concentration using a Qubit system. All sequencing data have been deposited under NCBI accession number PRJNA604975.

**Bioinformatics.** De novo assembly was performed using Shovill (v1.0.4)[20]. Reads were mapped to sequence type (ST[21,22]) specific references using Snippy (v4.6.0)[23] (Table S1). For the four *E. coli* STs with the largest number of isolates (i.e., *E. coli* ST131/95/73/69), we created core genome alignments using Snippy-core with the –mask auto setting, padded with the reference base at invariant positions; these whole genome alignments were used as input to Gubbins (v2.3.4)[24]. Such recombination-corrected phylogenies were also created for *E.hormaechei* (the most common non-*E. coli*/*Klebsiella* species detected in our study) and *S. marcescens* (because there was thought to have been an outbreak in our neonatal intensive care unit in 2016). We also used genomic distances calculated by Mash[25] (using -k21 -s 100,000 for within-study comparison of isolates and -k21 -s 1000 for comparison of isolates in this study to our collection of sequences of adult bloodstream infection isolates[26] from the same region over the same time period due to computational feasibility). Annotation against reference databases (VFDB/ResFinder) was performed using ABRicate (v2.3.4)[27] with genes called as being present if there was ≥80% coverage and DNA identity compared to the reference. Sequence types were predicted using the MLST tool (v2.19)[28]. For *Klebsiella* spp. isolates speciation and virulence gene detection (Supplementary Methods) was performed using Kleborate (v2.0.4)[29]. Detailed QC metrics and raw Abricate/Kleborate output have been uploaded to Figshare (https://figshare.com/projects/Paediatric_GNBSIs_in_Oxfordshire/135254).

**Definitions.** We defined isolates as being likely of neonatal origin if they originated from infants (i) in their first 30 days of life, or were (ii) in the neonatal intensive care unit, or (iii) under the care of a neonatologist on the day the blood culture was taken. For analytical purposes, we classified other children as <12 months, 1–4 years, 5–9 years and 10–17 years of age. Early-onset infection was defined as a disease within the first 72 h of life[30]. We further categorised BSIs according to healthcare exposure prior to onset as follows: nosocomial (>48 h after admission to hospital), 'quasi-nosocomial' (within 30 days of last discharge), 'quasi-community'

(31–365 days since last discharge) and community (>365 days since last discharge)[31]. Genetic relatedness in the form of single nucleotide polymorphism (SNP) thresholds definitively associated with the transmission is variably defined for the species evaluated; based on recent studies, we considered a threshold of >20 SNVs between isolates as highly unlikely to be representative of a transmission event[32].

**Epidemiology/statistics.** Routinely collected healthcare data were acquired via pseudonymised linkage in the Infections in Oxfordshire Research Database (IORD). IORD has generic Research Ethics Committee, Health Research Authority and Confidentiality Advisory Group approvals (19/SC/0403, 19/CAG/0144) as a de-identified electronic research database. Data on suspected infectious focus (only available for *E. coli/Klebsiella* spp.) were acquired via linked local infection control records, which had been submitted to Public Health England as part of the mandatory surveillance programme. Likely source was identified by infectious disease/microbiology physicians using best clinical judgement or designated as an unknown where there was uncertainty. For each species, we modelled the number of bloodstream infections (BSIs) per year using negative binomial regression, with the total number of paediatric admissions in each year used as an offset to account for changes in the population over time. Only complete years (i.e., excluding 2008 and 2018) were included in this part of the analysis. All statistical analysis was performed in R v4.0.3[33,34].

**Ethics statement.** The Infections in Oxfordshire Research Database (IORD; https://oxfordbrc.nihr.ac.uk/research-themes-overview/antimicrobial-resistance-and-modernising-microbiology/infections-in-oxfordshire-research-database-iord/) has generic Research Ethics Committee, Health Research Authority and Confidentiality Advisory Group approvals (19/SC/0403, 19/CAG/0144) which facilitate the pseudo-anonymised linkage of routinely collected NHS electronic healthcare record data from the Oxford University Hospitals NHS Foundation Trust Clinical Systems Data Warehouse and research data (e.g., sequencing data) from the Antimicrobial Resistance and Modernising Microbiology Theme of the Oxford NIHR Biomedical Research Centre, Oxford. IORD links records by a specific, random number ensuring that no patient-identifiable information is shared with researchers using this resource. Individual informed consent is not required under these permissions, which allow the lawful collection, storage and use of this data as a 'Public Task' under GDPR; individuals can opt-out of having their data included should they wish. Further details are available at https://oxfordbrc.nihr.ac.uk/wp-content/uploads/2020/01/IORD-privacy-note-2019-10-29.pdf. We sequenced bacterial isolates from bloodstream infections that are routinely stored by the John Radcliffe Hospital Microbiology laboratory. In the UK, bacterial isolates (such as those sequenced in this study) routinely cultured from human clinical samples do not require ethical approval for analysis under the provisions of the Human Tissue Act as they do not contain any material considered to be human tissue.

**Reporting summary.** Further information on research design is available in the Nature Research Reporting Summary linked to this article.

## Results

**Microbiology of neonatal and paediatric Gram-negative bloodstream infections in Oxfordshire.** Of the 327 GNBSI isolates cultured during the study period from individuals aged <18 years, 149 (46%) were identified as *E. coli* and 69 (21%) as *Klebsiella* spp.; the remaining 109 (33%) belonged to other species (n.b. latter

category only collected from 2011). There was no evidence of a change in the incidence of *E. coli* bloodstream infections (BSIs) either overall (incidence rate ratio per year [i.e., the relative increase/decrease in incidence per year] IRRy: 0.96, 95%CI: 0.90–1.03, $p = 0.30$) or in neonates (IRRy: 0.97, 95%CI: 0.86–1.06, $p = 0.39$). This was also the case for other Gram-negative species (i.e., non-*E. coli* and *Klebsiella* spp.) both overall (IRRy: 1.07, 95% CI: 0.93–1.24, $p = 0.33$) and in the neonatal group (IRRy: 0.83, 95% CI: 0.65–1.07, $p = 0.19$). Conversely, there was some evidence of a decrease in the overall number of *Klebsiella* spp. BSIs in the same period (overall IRRy: 0.91, 95%CI 0.83–1.00, $p = 0.06$; neonates IRRy: 0.80, 95%CI: 0.69–0.92, $p = 0.002$).

Neonates had the highest number of GNBSI (124/327 [38%], of which eight were early-onset) followed by other infants <1 y (66/327 [20%]) (Supplementary Data File 1). There were 62/327 (19%) cases in the 1st–4th years of life, 28/327 (9%) in the 5th–9th and 47/327 (14%) in the 10th–17th. A higher proportion of isolates came from males than would be expected from the birth:sex ratio (201/327 [61%], multinomial goodness of fit $p < 0.001$). BSIs in neonates, children aged 1–4 and 10–17 were predominantly nosocomial or quasi-nosocomial, whereas, in those aged 3–12 months and 5–9 years, they were predominantly community or quasi-community onset (Supplementary Data File 1).

In our network of hospitals, amoxicillin and gentamicin are used as empirical treatments for neonatal sepsis. In this patient group, 87/119 (73%) isolates were resistant to amoxicillin and 10/122 (8%) to gentamicin (missing data for five and two isolates, respectively); 7/118 (6%) isolates (4/56 *E. coli* and 3/27 *Klebsiella* spp.) were resistant to both. In older children (>1 month), ceftriaxone is the empirical agent used in our setting and in this population 24/183 (13%) isolates were resistant (missing data for 20 isolates). The proportion of resistant isolates appeared broadly stable over time (Fig. 1/S1). For gentamicin, however, there was some evidence that the proportion of susceptible isolates with a MIC >1 mg/L (IRRy 1.86, 95%CI 1.33–2.58) increased compared to those with a MIC ≤1 mg/L (IRRy 1.13, 95%CI 1.04–1.22) $p_{heterogeneity} = 0.002$ (EUCAST gentamicin breakpoint for resistance >2 mg/L). There was no evidence of substantial differences in phenotypic profiles between treating specialities (Fig. S2). Likewise, the proportion of resistant isolates was similar for nosocomial vs. community-onset cases for all agents except amoxicillin, where nosocomial isolates were proportionally more resistant (99/118 (84%) vs 115/174 (66%), Table S2), reflecting the higher burden of *Klebsiella*/*Enterobacter* spp. BSIs in this patient group (77/126 (61%) vs 101/201 (50%); both these genera are normally considered intrinsically resistant to amoxicillin).

Data on likely primary infective focus was only available for 78 isolates (66 *E. coli*/12 *Klebsiella* spp.), of these, 18 (23%) were of gastrointestinal origin, 8 (10%) from invasive lines, 20 (26%) from the urinary tract, 2 (3%) from the chest and 30 (38%) from unclear/other sources. Seventeen children (six neonates) had infections with >1 species and five (two neonates) had polyclonal infections (>1 ST of the same species).

**Molecular epidemiology of *Escherichia coli* causing BSI, including AMR and virulence gene burden.** Of the 149 *E. coli* isolates cultured, 133 (87%) were successfully retrieved for sequencing (Fig. 2). The four dominant *E. coli* STs (ST131 $n = 16$, ST95 $n = 16$, ST73 $n = 20$ and ST 69 $n = 15$) were the same as those in adults[26]. There was no evidence that the population structure observed differed from that in the adult population in Oxfordshire (which is also the same as that observed globally[26]; multinomial goodness of fit: $p = 0.44$). The most prevalent O-antigen types were O6 ($n = 21$), O1 ($n = 14$), O2 ($n = 12$), O16

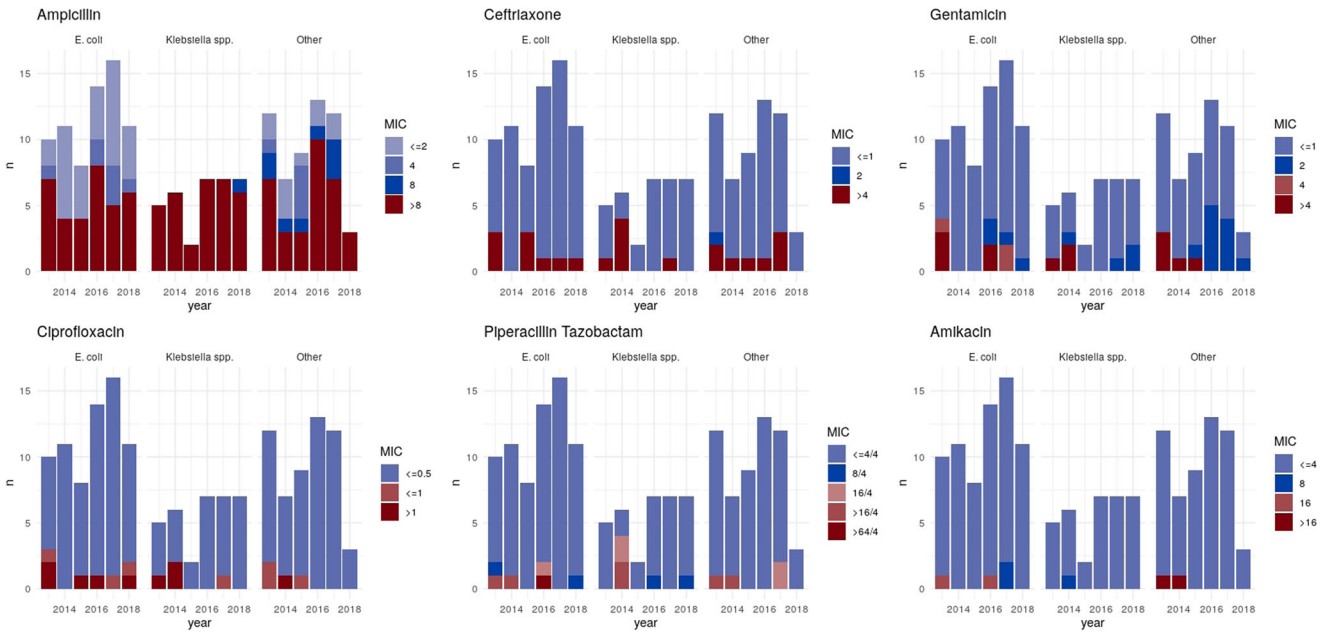

**Fig. 1 Change in minimum inhibitory concentrations (MIC) of isolates over time.** MIC is shown for each drug in mg/L.

($n = 10$) and O25 ($n = 8$). The proportion of isolates with an O-antigen included in the ExPEC-4V vaccine[35,36] (which has been evaluated in a phase-II study in adults) was similar when comparing all children (55/133, 41%) and neonates (25/53, 47%) to that seen in the overall population in our previous study[14] (1499/3278 (46%); chi-squared: $p = 0.4$ for both).

Carriage of genes conferring resistance to ampicillin was common (65/133, 49% isolates carrying 82 alleles), with $bla_{TEM-1B}$ (37/82, 45%) being the most prevalent allele. Eight isolates carried a gentamicin resistance gene (5 $aac(3)$-IIa [3 in ST131, 1 each in STs 73 and 23] and 3 $aac(3)$-IId [1 each in STs 95, 131 and 69]) and 18/133 (14%) carried a gene conferring resistance to ceftriaxone (mostly ESBL genes, namely $bla_{CTX-M-15}$ [$n = 9$], $bla_{CTX-M-14b}$ [$n = 3$] and $bla_{SHV-102}$ [$n = 5$]); Fig. 3). Whilst ST131 was the dominant ESBL-gene-carrying ST (7/16, 44%), other STs carrying these genes included ST73 (4/20), ST2141 (3/3), ST12 (2/7) and STs 23 and 38 (1 each).

**Molecular epidemiology of *Klebsiella* spp. causing BSI, including AMR and virulence gene burden.** Of the 69 *Klebsiella* spp. isolates cultured, 60 (87%) were successfully retrieved for sequencing. The predominant *Klebsiella* spp. in both neonates and older infants/children was *Klebsiella pneumoniae* ($n = 28/60$ [47%]), though a diverse selection of related species were also occasionally isolated (*K. oxytoca*: $n = 12$ [20%], *K. michiganensis*: $n = 8$ (13%), *K. aerogenes*: 5 (8%), *K. grimontii*: $n = 3$ (5%), *K. ornithinolytica*: $n = 2$ (3%) and *K. variicola*: $n = 2$ (3%)). As we have previously observed in adults and in contrast to the more clonal population seen in *E. coli*, STs causing *Klebsiella* spp. BSI were diverse ($n = 32$ STs in total). There were 11 isolates to which no ST could be assigned. We recently observed that four O-antigens (O2v2, O1v1, O3b and O1v2) were found in 75% of isolates from all *K. pneumoniae* BSIs in Oxfordshire over a 10-year period; amongst paediatric BSIs, this figure was 61% (17/28; Fisher's exact test: $p = 0.15$).

Half of *Klebsiella* spp. isolates (30/60) carried the yersiniabactin virulence factor (*ybt*); these were mostly non-*K. pneumoniae* (12/12 *K. oxytoca*, 7/8 *K. michiganensis*, 2/2 *K. ornitholytica* and 3/3 *K. grimontii*). One *K. aerogenes* isolate additionally carried the genotoxin colibactin (*clb*). There was no difference in the

proportion of E. coli/Klebsiella spp. isolates carrying a gene conferring ceftriaxone resistance (18/133, 14% vs 7/60, 12%, Fisher's exact test $p = 0.82$). Only 4/60 (7%) isolates carried a gentamicin resistance gene (four $aac(3)$-IIa and one $aac(3)$-Ia, similar to E. coli (8/133, 6%); Fisher's exact test: $p = 0.90$).

**Other Gram-negative species.** Of the 104 non-*E. coli/Klebsiella* spp. ('other') GNB cultured in this study, 77 (74%) were successfully sequenced. The predominant species in both neonates and older children was *E. hormaechei* (overall: 34/75, [44%]; neonatal: 14/25 [56%]). This was also the only other species carrying ESBL-producing genes ($bla_{CTX-M-15}$ in four isolates, Fig. 3); these isolates also carried the $bla_{TEM-1B}$ beta-lactamase as well as $aac(3)$-IIa conferring gentamicin resistance. Two isolates carrying class-D beta-lactamases predicted to confer carbapenem resistance were identified ($bla_{OXA-23}$ in *Acinetobacter baumannii*, $bla_{OXA-427}$ in *Aeromonas caviae*).

**Genomic relatedness of isolates.** Analysis of recombination-corrected phylogenies for the major *E. coli* STs (131/95/69) demonstrated no clear evidence of direct or temporally related indirect transmission between patients (with the closest genomic distance between patients = 32/87/28 single nucleotide variants [SNV] for ST131/95/69 respectively). There was a cluster of three near-identical ST73 isolates from two patients (3/6 SNPs apart); however, the isolation date was ~2 years apart, and these patients had never been admitted to the hospital at the same time. There was a cluster of three multidrug-resistant *E. hormaechei* isolates separated by a maximum of 37 SNPs. All three patients had been admitted in the same year but not at the same time and to different wards. Other *E. hormaechei* isolates were genomically diverse (Fig. 4). Three *S. marcescens* from the neonatal ICU in 2016 were grouped together in the phylogeny (Fig. S3) however, the relatively large distances between them (31/81/85 SNPs), made recent transmission unlikely.

There was no difference in the distributions of genomic distances (reflected here by Mash distances) between isolates from patients admitted (i) in the same year, (ii) to the same ward in the same year, or (iii) to the same ward with overlapping admission dates and (iv) overall (i.e., all admissions to any ward

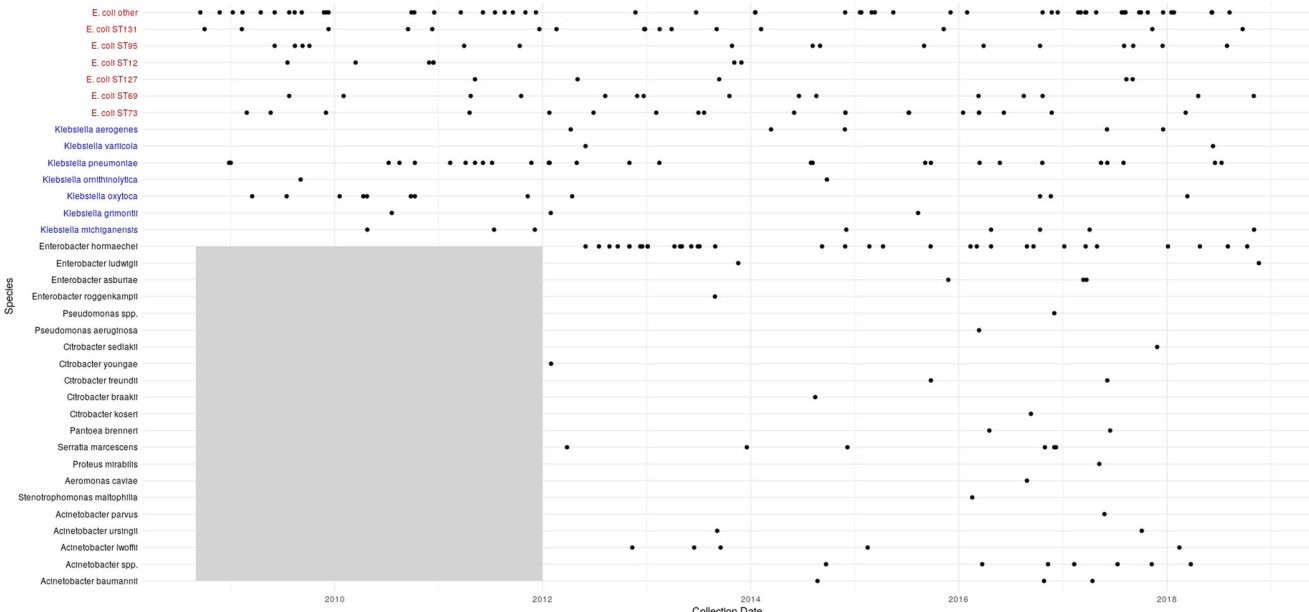

**Fig. 2 Species (and sequence type, ST, where shown) for sequenced isolate over time.** Each point represents the isolation date of a sequenced isolate. The grey shaded area represents the fact that sequencing of non-*E. coli/Klebsiella* spp. isolates did not begin prior to 2012.

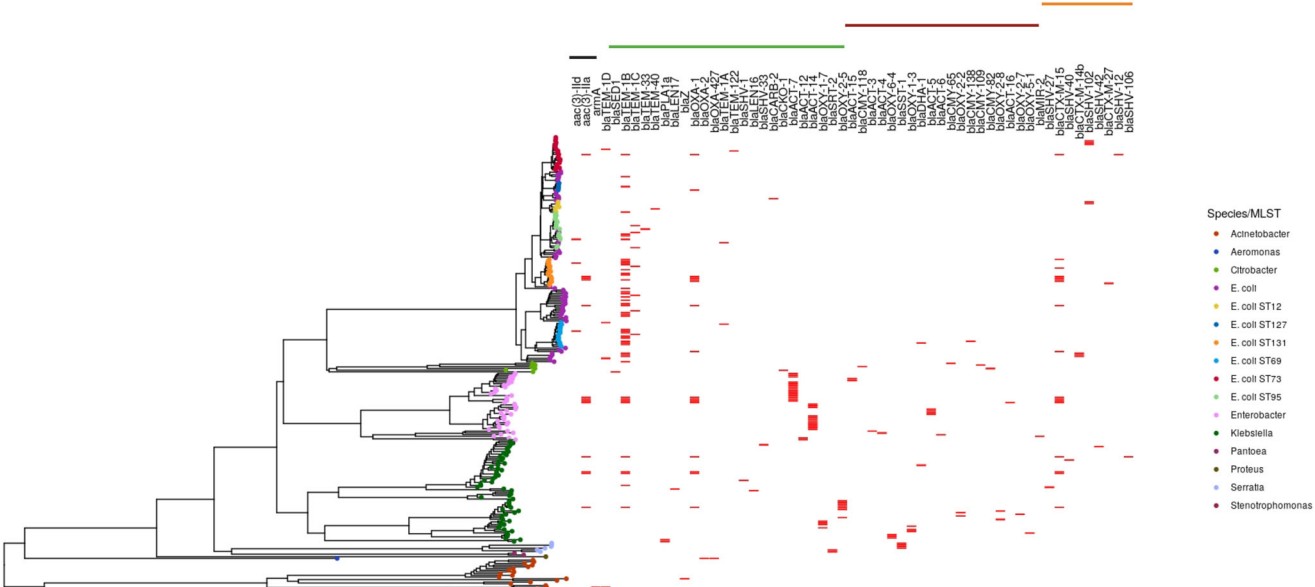

**Fig. 3 Carriage of antimicrobial resistance genes by isolates sequenced in the study.** The tree was created using mashtree and gene presence is shown in red. The black/green/maroon/orange bars at the top represent the carriage of genes producing enzymes with activity (as predicted by Resfinder) against aminoglycosides/amoxicillin/cefotaxime (+amoxicillin)/ceftriaxone (+cefotaxime/amoxicillin) respectively.

at any time point) (Kruskal–Wallis rank-sum test $p = 0.30$, Fig. S4). Finally, when considering *E. coli* and *Klebsiella* isolates in this study, there was no difference between the Mash distances to the nearest paediatric genomic neighbour (median: 0.00093 [IQR: 0.00036–0.0023]) compared to the nearest adult neighbour (median: 0.0011 [IQR: 0.00041–0.0024], Wilcoxon rank-sum $p = 0.33$. Fig. S5).

## Discussion

*E. coli* was the main causative agent of paediatric and neonatal Gram-negative bloodstream infection in our study over a 10-year period. The population structure of *E. coli* isolates causing invasive infection in children mirrors that seen in our centre and globally for adults. We found no evidence of substantial nosocomial transmission of isolates causing bloodstream infection, suggesting that in our setting, invasive isolates are acquired from the environment, represent colonising gastrointestinal flora[37], or as a result of transmission from parents/other close contacts. Our findings suggest O-antigen targeted vaccines (currently in phase-II/III trials in adults) might have the potential to reduce the incidence of invasive neonatal and paediatric *E. coli* infections, in the former potentially by immunisation of pregnant women. Most isolates remained susceptible to at least one first-line

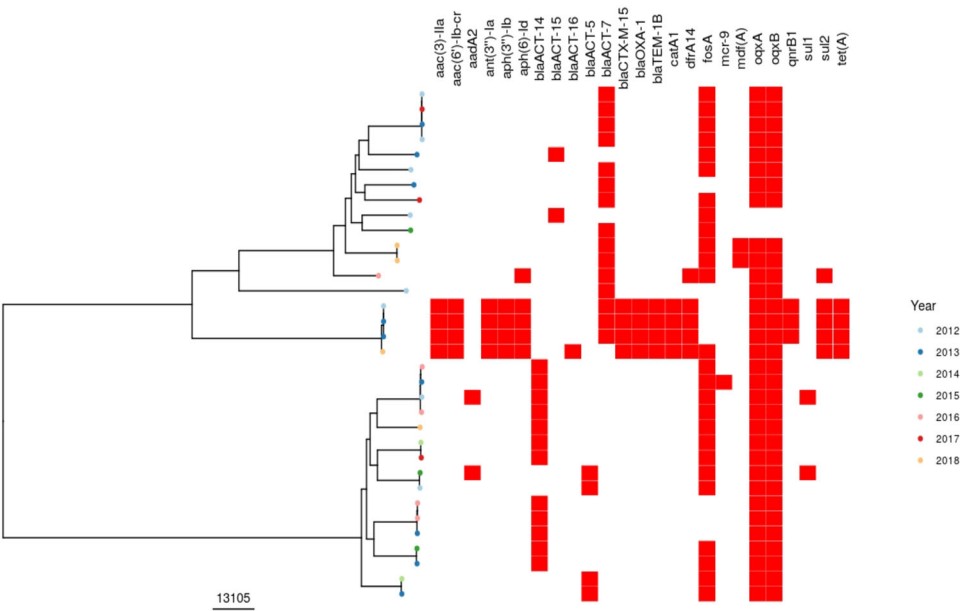

**Fig. 4 Recombination-corrected phylogeny of *Enterobacter hormaechei* isolates sequenced in this study.** Tree tip colours show the year of isolates, red bars denote the presence/absence of antimicrobial-resistance genes. The scale of the tree is shown in SNPs.

antibiotic justifying the continuation of current empirical treatment guidelines, which should include an aminoglycoside in neonates[30]. There was, however, some evidence of increasing (sub-breakpoint) gentamicin MIC, which warrants ongoing surveillance, and 1 in 20 GNBSIs were caused by isolates resistant to the ampicillin + gentamicin combination.

The incidence of these infections has remained stable in Oxfordshire over the past decade in children, in contrast to the increase observed in adults. The substantial burden of disease due to paediatric GNBSI occurs in neonates. The absence of evidence of an increasing burden of AMR-associated disease and demonstrable outbreaks likely points to good infection control practice and antimicrobial stewardship. Continued epidemiological surveillance with targeted whole genome sequencing to identify the strain and/or plasmid outbreaks may help to inform early infection control interventions and would seem particularly justified in these age groups given the high associated morbidity and mortality. In addition, we would encourage consideration of clinical trials of existing O-antigen targeted vaccines for the prevention of neonatal sepsis.

Our data highlight the importance of *E. hormachei* as a potential causative agent of neonatal and paediatric sepsis as well as its potential to cause multidrug-resistant disease in this setting. Several small clusters of *E. hormaechei* (part of the *E. cloacae* complex) bloodstream infection in neonates have been described in the literature, with enteral feeding and milk formulations as well as poor general infection control procedures thought to be implicated[38–40].

Our data highlights the value of whole genome sequencing for the investigation of potential outbreaks in neonatal units. WGS analysis of three isolates from a suspected (on the basis of epidemiology and pulsed-field gel electrophoresis) *S. marcescens* outbreak suggested recent transmission was actually highly unlikely. Had this analysis been performed in real-time, it may have reassured clinicians and saved time and resource. Apart from a small cluster of three patients with *E. hormaechei* infections where acquisition from a common source was possible based on genomic and epidemiological analysis (though the patients were never admitted to the same ward), there was relatively high diversity amongst isolates, effectively ruling out transmission/point-source acquisition resulting in BSI. However,

unwell neonates are often transferred across a geographic region for specialist care, and so it is essential that active surveillance mechanisms (collecting and analysing real-time microbiological data) are implemented across networks in order to inform infection prevention measures appropriately.

Notably, whilst we found a lack of clear evidence of transmission between patients in this study, BSIs are likely to represent an extremely insensitive marker of transmission events, as most transmission with these organisms would be expected to lead to colonisation and not necessarily invasive infection. Furthermore, whilst the short-read sequencing used in this study allows us to confidently exclude outbreaks of strains, we cannot address the question of potential transmission of AMR genes and virulence factors on plasmids. Additional limitations include the fact that we were not able to sequence all cultured isolates, the relatively sparse availability of data on source attribution and the single region nature of the study. We did not sample other reservoirs potentially contributing to transmission, including the unit environment.

In summary, our study demonstrates a flat overall incidence trend and stable numbers of AMR-carrying GNBSI isolates in children, contrasting with what has been seen in adult populations in the same setting[26]. Pertinently our data support the ongoing use of an aminoglycoside as part of empirical treatment guidelines in neonatal sepsis. The disproportionate impact of GNBSI on neonates and young infants should encourage active microbiological surveillance across neonatal networks and strategies to prevent disease through vaccine trials in pregnant women.

## Data availability

All sequencing data have been deposited under NCBI accession number PRJNA604975. Raw metadata can be obtained by accredited researchers by making an application to IORD (see https://oxfordbrc.nihr.ac.uk/research-themes-overview/antimicrobial-resistance-and-modernising-microbiology/infections-in-oxfordshire-research-database-iord/). Source data used to create the figures has been deposited at https://doi.org/10.6084/m9.figshare.20330862.v1.

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

## Acknowledgements

The research was supported by the National Institute for Health Research (NIHR) Health Protection Research Unit in Healthcare Associated Infections and Antimicrobial Resistance (NIHR200915) at the University of Oxford in partnership with Public Health England (PHE) and by Oxford NIHR Biomedical Research Centre. T Peto and AS Walker are NIHR Senior Investigators. The report presents independent research funded by NIHR. The views expressed in this publication are those of the authors and not necessarily those of the NHS, NIHR, the Department of Health or Public Health England. The computational aspects of this research were funded by the NIHR Oxford BRC with additional support from the Wellcome Trust Core Award Grant Number 203141/Z/16/Z. S.L. is supported by a Medical Research Council Clinical Research Training Fellowship. K.C. is Medical Research Foundation-funded.

## Author contributions

The study was conceived by S.L., N.S., and S.K. K.J., S.P., L.B., S.O., L.B., and S.S. provided access to data. S.L. wrote the first draft of the manuscript and led the analysis. S.Wr., M.T., S.S., S.P., L.B., S.O., K.J., and Li.B. were involved in data curation. T.D., K.C., A.V., J.K., L.B., and S.G. performed the laboratory work. K.-D.V. provided statistical advice. T.P., D.C., S.W., S.K., and N.S. supervised the project. All authors reviewed and provided feedback on the final version of the manuscript.

## Competing interests

The authors declare no competing interests
