## [Peer Review File · Communications Medicine]

Reviewers' comments:

Reviewer #1 (Remarks to the Author):

In their manuscript “Molecular epidemiology and antimicrobial resistance phenotype of paediatric bloodstream infections caused by Gram-negative bacteria in Oxfordshire, UK”, the authors report a molecular epidemiological analysis of gram-negative isolates causing bloodstream infections over a ten-year period in a pediatric population in Oxfordshire, England. Illumina short-read whole genome sequencing was successfully performed on 296 isolates allowing for sequence typing, core genome alignment, and detection of candidate virulence and antimicrobial resistance-conferring genes. Phenotypic antimicrobial susceptibility information from these isolates was used to provide additional insight into the burden of resistance to antibiotics commonly used in the studied population. By analyzing the relatedness of isolates across space and time, the authors state that it is unlikely transmission events occurred during the isolate collection period and that the sequence types of Gram-negative isolates from the pediatric population studied did not differ from those observed either in adults in the same health system or globally. The authors claim that isolates from younger patients, including neonates, tend to have a higher burden of virulence genes than older patients, however that the isolates from older patients have a higher antimicrobial resistance score on average.

Overall, this paper provides a thorough sequencing-based and phenotypic analysis of isolates causing bloodstream infections in a pediatric population, which has previously been an understudied area despite neonates and young children being vulnerable to serious bacterial infections. The isolates are numerous and span an extended timeframe which allows for understanding possible changes in antimicrobial resistance or causative agents over time. This paper provides evidence for the continuation of current empirical guidelines for treating gram-negative bacterial infections in children and suggests a lack of transmission between patients included in the study, refuting a previously suspected outbreak of *Serratia marcescens* in the neonatal ICU. However, although the findings would incrementally add to the body of evidence, there are important concerns about the methodology and presentation of the data in the current form of manuscript that would need to be addressed before it should be considered for publication. Major and minor concerns are noted below:

Major concerns:

1. It is not stated why 31 of the included isolates failed sequencing. At what stage did these samples drop out and were other methods attempted to obtain sequencing data? It is also not stated what genome coverage and read count was achieved for the whole genome sequencing. In order to perform resistance and virulence gene analysis, it is important to ensure at least 75x genome coverage so the genome assemblies are valid. Please also present genome assembly statistics including completeness, contamination, N50, etc. Programs such as quast and checkM would enable this data to be presented.
2. The main figures do not appropriately aid the reader in interpreting the data, and much of the results discussed in the text are not shown graphically or pictorially. For example, the bar graphs over time in Figure 1 are not informative. All the figures are reported by year of isolation, but as there was no impact of time on metrics reported, this confuses the reader. Furthermore, quantification of MIC testing is inferred and would require the reader to calculate numbers of susceptible or resistant isolates. It would be preferred to include boxplots of other graphical representations of isolates and their MICs. It may be helpful to see Flannery et al. 2020 or 2021 for examples of how to clearly present the type of data included in this manuscript.

3. It is unclear what conclusion is being drawn from the observation that the virulence factor count is highest in younger children and neonates, while the resistance score increases. The data supporting this point is presented in Figure S2 but it is not stated how these values were determined. It is unclear which virulence factors are being included and their biological significance. If this is being presented as a major finding it should be in the main figures and there should be careful discussion about the potential reasons why these values correlate with age within their population. Furthermore, ABRicate and Kleborate should be rerun at 90% and 95% coverage since 80% or greater coverage may not be specific enough to call genes. Additionally, it is possible that the same gene is double counted for certain isolates given this low threshold.

4. Table 1: The authors reference Fisher's exact test and Kruskal-Wallis; however, for species and age this would be categorical and more than a 2x2 table therefore Chi Square analysis would be necessary. Similarly, for later metrics, it is unclear to what the p value refers. For example, for antibiotic susceptibility results, does the p value refer to the difference between susceptible and resistant across all time cohorts?

Minor concerns:

5. Lines 83-87: Were recurrent bacteremia isolates from the same patient excluded if they matched morphotypes?
6. Lines 83-87: Why were other GNB not included from 2008-2011?
7. Line 107: should "ITU" be "ICU"?
8. Table 1: Age should have (days or d) to clarify that all ages are in days of life
9. Table 1: It is not a fair comparison to make between neutrophils and creatinine between different age groups as the normal values differ (see <https://www.ncbi.nlm.nih.gov/pmc/articles/PMC7462212/> for example). Please remove these rows also as no reference to them is made in the text.
10. Figure S1 does not add value to the manuscript and should be removed.
11. Lines 178-180: Table S1 does not report antimicrobial susceptibility testing as described. Also, it does not necessarily follow that non-E. coli BSI would, by definition, be more likely resistant to amoxicillin. Please revise.
12. Figures 1 and 2 are inverted in the Figure legends.
13. Figure 3: how was the distinction made between genes encoding cefotaxime versus ceftriaxone resistance?
14. Line 248: "20116" should be 2016
15. Line 264-266: From within the gut should be listed as another mechanism of bloodstream invasion (see for example Tamburini et al 2018 Nature Medicine or Carl et al 2016 JID)
16. Line 288-290: "this" pathogen should be "these pathogens" as it refers to non-E. coli/kleb species or revise to focus exclusively on E. hormachei. Similarly, data is not presented of these isolates being multidrug-resistant so please remove this or discuss in the references the quantity that are multidrug-resistant.

Reviewer #2 (Remarks to the Author):

Molecular epidemiology and antimicrobial resistance 1 phenotype of paediatric bloodstream infections caused by Gram-negative bacteria in Oxfordshire, UK

Lipworth et al
28th June 2022

Adam Irwin

General comments and recommendation:

The manuscript is a very well-presented report on the molecular epidemiology of Gram-negative bloodstream infection in children in Oxfordshire over more than a decade. It addresses an important topic of public health interest for which presently only limited published data exist. It was interesting and clear to read.

Major comments:

Overall:

I think the manuscript provides a valuable insight into the molecular epidemiology of these important infections in children. Limitations of the analysis (such as the nature of the accompanying clinical information, and the absence of long-read sequencing to support the identification of plasmids) are appropriately addressed by the authors in the discussion. I would suggest the raw data upon which the analysis of temporal trends is performed are available as a supplementary table (number of isolates by species by year).

Abstract:

The abstract is clear and compelling and presents the key findings.

Introduction:

The introduction sets the scene well. The final paragraph makes a number of assertions that ought to be cited. For example, line 66 referring to the 'emergence of particular AMR-associated sequence types' could reference the paper by Petty et al which reported this (Global dissemination of a multidrug resistant *Escherichia coli* clone. PNAS 111 (15) 5694-5699; doi.org/10.1073/pnas.1322678111).

Methods:

The methods are clear. The authors don't explain why data are reported on *E coli* and *Klebsiella* from 2008, but from 2011 for other Gram negatives. The bioinformatics methods appear appropriate, but I would defer to a specialist.

Line 145. An appropriate citation should be given for R, and ideally the packages used. Use citation () function.

Reference is made to the AMR score, but not a virulence gene score. Was a formal method used for this, and how does it relate to the results presented on virulence gene carriage (eg line 207)?

Results:

The crude proportions make no reference to the fact that data on *E coli* and *Klebsiella* were collected for a longer period.

The first paragraph reported IRR for *E coli*, then 'other Gram-negative species' before concluding with *Klebsiella*. This gives the impression (I suspect mistakenly) that the 'other Gram-negative' category includes *Klebsiella* (as it follows on directly from *E coli*). Consider reporting *E coli*, then

Klebsiella then 'Other Gram-negatives)

Line 156 'slight' seems redundant. The analysis suggests a reduction of ~10% per year in children and 20% in neonates.

Would a population-based denominator be more informative for presentation of the relative incidence by age?

I found it hard to relate the proportions of resistant isolates in the paragraph starting at line 167 to the paragraph above which gave the breakdown by age. For example, which age group does 'older children (> 1 month)' relate to? 123 isolates are reported (24/103 resistant to CRO, and 20 missing), but I can't work out how this figure was arrived at if there are 66 infants > 1month, 62 aged 1-4y, 28 aged 5-9y and 47 >10y.

Line 224 which compares proportion of E coli/Klebsiella carrying a gene conferring resistance to CRO reports 19/137 relevant E coli, but this is 20/133 in the E coli section above (line 201).

Discussion:

Line 275. It would be worth relating this statement to Oxfordshire.

Line 287 this assertion doesn't seem to be supported by the results presented.

Line 318 'Relative success story' seems an odd turn of phrase for an observational study.

I found the bars at the top of Figure 3 'representing the carriage of genes producing enzymes with activity against...' very confusing. While Figure 3 presents a great deal of data, I wonder if the presentation of all species on the single tree risks obscuring important information.

Minor comments:

I would remove 'sepsis' from the running title. Sepsis is really a different clinical phenomenon not specifically relevant to the data presented.

Presenting age in days in Table 1 is messy.

Dear Editor,

Thankyou for considering our manuscript. We have taken time to carefully consider the points of the reviewers below and enclose a point-by-point response. For ease of reading we have provided narrative responses in blue and changes to the manuscript in red.

Kind Regards,
Sam Lipworth

Reviewer #1 (Remarks to the Author):

In their manuscript “Molecular epidemiology and antimicrobial resistance phenotype of paediatric bloodstream infections caused by Gram-negative bacteria in Oxfordshire, UK”, the authors report a molecular epidemiological analysis of gram-negative isolates causing bloodstream infections over a ten-year period in a pediatric population in Oxfordshire, England. Illumina short-read whole genome sequencing was successfully performed on 296 isolates allowing for sequence typing, core genome alignment, and detection of candidate virulence and antimicrobial resistance-conferring genes. Phenotypic antimicrobial susceptibility information from these isolates was used to provide additional insight into the burden of resistance to antibiotics commonly used in the studied population. By analyzing the relatedness of isolates across space and time, the authors state that it is unlikely transmission events occurred during the isolate collection period and that the sequence types of Gram-negative isolates from the pediatric population studied did not differ from those observed either in adults in the same health system or globally. The authors claim that isolates from younger patients, including neonates, tend to have a higher burden of virulence genes than older patients, however that the isolates from older patients have a higher antimicrobial resistance score on average.

Overall, this paper provides a thorough sequencing-based and phenotypic analysis of isolates causing bloodstream infections in a pediatric population, which has previously been an understudied area despite neonates and young children being vulnerable to serious bacterial infections. The isolates are numerous and span an extended timeframe which allows for understanding possible changes in antimicrobial resistance or causative agents over time. This paper provides evidence for the continuation of current empirical guidelines for treating gram-negative bacterial infections in children and suggests a lack of transmission between patients included in the study, refuting a previously suspected outbreak of *Serratia marcescens* in the neonatal ICU. However, although the findings would incrementally add to the body of evidence, there are important concerns about the methodology and presentation of the data in the current form of manuscript that would need to be addressed before it should be considered for publication. Major and minor concerns are noted below:

Thankyou for taking the time to read and review our manuscript.

Major concerns:

1. It is not stated why 31 of the included isolates failed sequencing. At what stage did these samples drop out and were other methods attempted to obtain sequencing data?

These isolates could unfortunately not be found in the isolate archive. We have amended the text to state "...were successfully retrieved for sequencing" where relevant.

It is also not stated what genome coverage and read count was achieved for the whole genome sequencing. In order to perform resistance and virulence gene analysis, it is important to ensure at least 75x genome coverage so the genome assemblies are valid. Please also present genome assembly statistics including completeness, contamination, N50, etc. Programs such as quast and checkM would enable this data to be presented.

We have added basic sequencing quality and assembly statistics to our Figshare repository (https://figshare.com/projects/Paediatric_GNBSIs_in_Oxfordshire/135254), demonstrating that our assemblies are of good quality and the isolates have been sequenced to a high depth (median 87X coverage, IQR: 74-103X). We have recently demonstrated that at ~15X coverage, several commonly used bioinformatic methods for resistance/virulence gene detection are able to correctly genotype isolates in >95% of cases, and there appears to be little benefit to increasing depth beyond that (see Figure 1 <https://doi.org/10.1101/2021.11.03.467004>).

2. The main figures do not appropriately aid the reader in interpreting the data, and much of the results discussed in the text are not shown graphically or pictorially. For example, the bar graphs over time in Figure 1 are not informative. All the figures are reported by year of isolation, but as there was no impact of time on metrics reported, this confuses the reader.

Whilst the Reviewer correctly points out that we found little change over time for most metrics analysed, this is in itself a key finding, accurately reflected in the bar graphs over time in Figure 1.

Furthermore, quantification of MIC testing is inferred and would require the reader to calculate numbers of susceptible or resistant isolates. It would be preferred to include boxplots of other graphical representations of isolates and their MICs. It may be helpful to see Flannery et al. 2020 or 2021 for examples of how to clearly present the type of data included in this manuscript.

The numbers of susceptible/resistant isolates are provided directly in table 1. We believe that presenting the data by MIC is informative for a clinical audience, but have also now provided the raw absolute counts by year on Figshare (https://figshare.com/projects/Paediatric_GNBSIs_in_Oxfordshire/135254) for interested readers to review, and a supplementary figure (Fig.S1) showing the proportion of non-susceptible isolates for each antibiotic over time as in Flannery D et al (<https://www.ncbi.nlm.nih.gov/pmc/articles/PMC7653538/>).

3. It is unclear what conclusion is being drawn from the observation that the virulence factor count is highest in younger children and neonates, while the resistance score increases. The data supporting this point is presented in Figure S2 but it is not stated how these values were determined. It is unclear which virulence factors are being included and their biological significance.

The methodology we used to calculate a resistance score is presented in the methods:

Lines 123-125: We calculated a 'resistance score' using a previously described method³¹ (the sum of the number of resistance gene categories carried out of amoxicillin, co-trimoxazole, cefotaxime, gentamicin and ciprofloxacin).

Similarly the methodology used for detecting virulence genes is presented in the methods:

Lines 110-112: Annotation against reference databases (VFDB/ResFinder) was performed using ABRicate (v2.3.4)²⁷ with genes called as being present if there was $\geq 80\%$ coverage and DNA identity compared to the reference.

We agree however that it is worth clarifying that we are referring to a count of the total number of detected genes (and not a score) and have clarified this in the relevant section:

Lines 158-160: Conversely there was some evidence that virulence gene carriage (number of such genes detected by Abricate in the VFDB) was higher in younger children (CMY = -0.53 95%CI -1.03 - -0.033, p=0.04 Figure S2).

If this is being presented as a major finding it should be in the main figures and there should be careful discussion about the potential reasons why these values correlate with age within their population.

We have added some discussion regarding the potential reasons these values correlate with age however any further elaboration would be pure speculation as the biology underpinning this finding (which has now been reproduced in two independent settings Burdet C, Clermont O, Bonacorsi S, et al. *Escherichia coli* bacteremia in children: age and portal of entry are the main predictors of severity. *Pediatr Infect Dis J.* 2014;33(8):872-879.) remains unclear:

Lines 285-289: ...in our setting we were able to replicate results from a previous study³¹ suggesting greater carriage of virulence genes in *E. coli* isolates from younger children and neonates but higher resistance scores in older children. The latter finding might be explained by increased hospital contact and environmental exposure in this group whereas reasons for and clinical significance of virulence gene acquisition is unclear.

Whilst we agree that this is potentially an interesting observation, we think adequate information to support the finding is presented in the main results and would prefer to keep this figure as supplementary, however are happy to be further directed by the editorial team on this.

Furthermore, ABRicate and Kleborate should be rerun at 90% and 95% coverage since 80% or greater coverage may not be specific enough to call genes. Additionally, it is possible that the same gene is double counted for certain isolates given this low threshold.

The selection of relevant thresholds for calling gene presence/absence and the methodologies for doing this are non-trivial and there is a sensitivity/specificity tradeoff as the Reviewer rightly points out (see e.g. <https://doi.org/10.1101/2021.11.03.467004>). It is true to say however that Abricate is a widely used tool and most comparable genomic epidemiology studies (e.g. <https://doi.org/10.1093/jac/dkab451>, <https://doi.org/10.1038/s41564-021-00870-7>) use it with default settings. This choice therefore ensures easy comparison with other similar studies. It is true to say that Abricate compared to other tools (e.g. Ariba) is biased in favour of longer matches which have more statistically significant scores and that this sometimes leads to erroneous calls, but Abricate does not call two genes for the same region, irrespective of threshold.

In response to the Reviewer's comments we have examined the coverage that Abricate reports for any gene which is included in our analysis – the minimum was 94.7%, median =100% (IQR 100% - 100%). We are therefore confident that a re-analysis using the suggested thresholds would be extremely unlikely to change the results that we present and yet would require a considerable amount of work. We recognise however that some readers may be interested in the fine details of resistance gene detection and have therefore uploaded the raw Abricate output to our Figshare repository.

4. Table 1: The authors reference Fisher's exact test and Kruskal-Wallis; however, for species and age this would be categorical and more than a 2x2 table therefore Chi Square analysis would be necessary.

Fisher's exact test can be run in R on tables that are larger than 2x2 using the generalisation described by Freeman-Halton (Freeman, G. H., and John H. Halton. "Note on an exact treatment of contingency, goodness of fit and other problems of significance." *Biometrika* 38.1/2 (1951): 141-149.). This extension is however still commonly referred to as Fisher's exact test and we think referring to it as the Freeman-Halton test would be confusing/unfamiliar for most readers.

Similarly, for later metrics, it is unclear to what the p value refers. For example, for antibiotic susceptibility results, does the p value refer to the difference between susceptible and resistant across all time cohorts?

We have clarified this in the table legend:

Table 1 – microbiological and patient characteristics of patients with Gram-negative bloodstream infections in Oxfordshire stratified by age of onset. P values represent row-wise comparisons using Fisher's exact tests for categorical and Kruskal-Wallis tests for continuous variables.

Minor concerns:

5. Lines 83-87: Were recurrent bacteremia isolates from the same patient excluded if they matched morphotypes?

Yes, if within the same 90-day period, as specified in the Methods (L88-89).

6. Lines 83-87: Why were other GNB not included from 2008-2011?

This was primarily a resource issue but as the problem of Gram-negative BSIs increased and sequencing costs dropped, we included the other isolates in the study. We have added a line to clarify this as follows:

Lines 86-87: The same selection criteria were applied to other GNB from August-2011 to September-2018 which were excluded from the initial period due to resource limitations.

7. Line 107: should "ITU" be "ICU"?

The terms are essentially interchangeable however we have now written it out in full to avoid confusion.

8. Table 1: Age should have (days or d) to clarify that all ages are in days of life.

Thankyou for noticing this omission which we have corrected.

9. Table 1: It is not a fair comparison to make between neutrophils and creatinine between different age groups as the normal values differ (see <https://www.ncbi.nlm.nih.gov/pmc/articles/PMC7462212/> for example). Please remove these rows also as no reference to them is made in the text.

We have removed these rows as suggested.

10. Figure S1 does not add value to the manuscript and should be removed.

We respectfully disagree and think that this is an interesting comparison which some clinicians would be interested in seeing.

11. Lines 178-180: Table S1 does not report antimicrobial susceptibility testing as described.

Also, it does not necessary follow that non-E. coli BSI would, by definition, be more likely resistant to amoxicillin. Please revise.

We apologise - we have now inserted the correct table S1 (which actually ought to be table S2). We have clarified that we meant that the higher proportion of amoxicillin resistant isolates in the nosocomial setting was likely caused by the higher proportion of *Klebsiella/Enterobacter* species (which are both normally considered intrinsically resistant to amoxicillin due to the expression of chromosomal beta-lactamases).

Lines 180-184: Likewise, the proportion of resistant isolates was similar for nosocomial vs. community-onset cases for all agents except amoxicillin where nosocomial isolates were proportionally more

resistant (99/118 (84%) vs 115/174 (66%), table S2), reflecting the higher burden of *Klebsiella/Enterobacter* spp. BSIs in this patient group (77/126 (61%) vs 101/201 (50%); both these genera normally considered intrinsically resistant to amoxicillin).

12. Figures 1 and 2 are inverted in the Figure legends.

Thanks - we have corrected this.

13. Figure 3: how was the distinction made between genes encoding cefotaxime versus ceftriaxone resistance?

This is purely based on what is reported by ResFinder (which we agree should be interpreted with caution particularly for *bla*_{SHV} type genes). We have clarified the legend as follows:

The black/green/maroon/orange bars at the top represent carriage of genes producing enzymes with activity (as predicted by ResFinder)

14. Line 248: "20116" should be 2016

Thankyou for spotting this typo - we have amended.

15. Line 264-266: From within the gut should be listed as another mechanism of bloodstream invasion (see for example Tamburini et al 2018 Nature Medicine or Carl et al 2016 JID)

We have clarified as follows and included the suggested reference.

Lines 269-272: We found no evidence of significant nosocomial transmission of isolates causing bloodstream infection, suggesting that in our setting invasive isolates are acquired from the environment, represent colonising gastrointestinal flora³⁸, or as a result of transmission from parents/other close contacts.

16. Line 288-290: "this" pathogen should be "these pathogens" as it refers to non-*E. coli*/*kleb* species or revise to focus exclusively on *E. hormachei*. Similarly, data is not presented of these isolates being multidrug-resistant so please remove this or discuss in the references the quantity that are multidrug-resistant.

Lines 235-237 discuss the multidrug-resistant nature of some *Enterobacter hormaechei* isolates in our setting and this is also shown on Figure 3. We have edited this section so that it only refers to *Enterobacter hormaechei*:

Lines 295-299: Our data highlight the importance of *E. hormachei* and as a potential causative agent of neonatal and paediatric sepsis as well as its potential to cause multidrug-resistant disease in this setting. Several small clusters of *E. hormaechei* (part of the *E. cloacae* complex) bloodstream infection in neonates have been described in the literature, with enteral feeding

and milk formulations as well as poor general infection control procedures thought to be implicated³⁹⁻⁴¹.

Reviewer #2 (Remarks to the Author):

Molecular epidemiology and antimicrobial resistance 1 phenotype of paediatric bloodstream infections caused by Gram-negative bacteria in Oxfordshire, UK

Lipworth et al
28th June 2022

Adam Irwin

General comments and recommendation:

The manuscript is a very well-presented report on the molecular epidemiology of Gram-negative bloodstream infection in children in Oxfordshire over more than a decade. It addresses an important topic of public health interest for which presently only limited published data exist. It was interesting and clear to read.

Thankyou for taking the time to read our manuscript, we are glad that you found it of interest.

Major comments:

Overall:

I think the manuscript provides a valuable insight into the molecular epidemiology of these important infections in children. Limitations of the analysis (such as the nature of the accompanying clinical information, and the absence of long-read sequencing to support the identification of plasmids) are appropriately addressed by the authors in the discussion. I would suggest the raw data upon which the analysis of temporal trends is performed are available as a supplementary table (number of isolates by species by year).

We have uploaded such a table of raw counts to Figshare (https://figshare.com/projects/Paediatric_GNBSIs_in_Oxfordshire/135254).

Abstract:

The abstract is clear and compelling and presents the key findings.

Introduction:

The introduction sets the scene well. The final paragraph makes a number of assertions that ought to be cited. For example, line 66 referring to the 'emergence of particular AMR-associated sequence types' could reference the paper by Petty et al which reported this (Global

dissemination of a multidrug resistant *Escherichia coli* clone. PNAS 111 (15) 5694-5699; doi.org/10.1073/pnas.1322678111).

We agree that citations in the final paragraph were omitted and have now ensured this section is properly referenced. Thankyou for the suggested citation which we have incorporated.

Methods:

The methods are clear. The authors don't explain why data are reported on *E coli* and *Klebsiella* from 2008, but from 2011 for other Gram negatives. The bioinformatics methods appear appropriate, but I would defer to a specialist.

Initially we were only resourced to sequence *E. coli* and *Klebsiella* spp. BSIs but as the incidence of all Gram-negative bloodstream infections continued to increase and the cost of sequencing fell, we expanded the scope to include the other species from 2011.

We have added a line to explain this:

Lines 86-87: "The same selection criteria were applied to other GNB from August-2011 to September-2018 which were excluded from the initial period due to resource limitations."

Line 145. An appropriate citation should be given for R, and ideally the packages used. Use citation () function.

We have added references as follows:

34. R Core Team. R: A Language and Environment for Statistical Computing. Published online 2019. <https://www.R-project.org/>
35. Wickham H, Averick M, Bryan J, et al. Welcome to the tidyverse. *Journal of Open Source Software*. 2019;4(43):1686. doi:10.21105/joss.01686

Reference is made to the AMR score, but not a virulence gene score. Was a formal method used for this, and how does it relate to the results presented on virulence gene carriage (eg line 207)?

We used number of genes in the virulence factor database (VFDB) as detected by Abricate rather than a score in line 207. We agree this was not clear and have clarified as follows:

Lines 211-213: "Conversely there was some evidence that virulence gene carriage (number of such genes detected by Abricate in the VFDB) was higher in younger children (CM_y = -0.53 95%CI -1.03 - -0.033, p=0.04 Figure S3)".

Results:

The crude proportions make no reference to the fact that data on *E. coli* and *Klebsiella* were collected for a longer period.

Whilst this is only intended as a high level description of what was sequenced, we agree that it might be misleading for readers who have not carefully read the methods. We have therefore inserted the following caveat/reminder for readers:

Lines 151-153: “Of the 327 GNBSI isolates cultured during the study period from individuals aged <18 years, 149 (46%) were identified as *E. coli* and 69 (21%) as *Klebsiella* spp.; the remaining 109 (33%) belonged to other species (n.b. latter category only collected from 2011). “

The first paragraph reported IRR for *E. coli*, then ‘other Gram-negative species’ before concluding with *Klebsiella*. This gives the impression (I suspect mistakenly) that the ‘other Gram-negative’ category includes *Klebsiella* (as it follows on directly from *E. coli*). Consider reporting *E. coli*, then *Klebsiella* then ‘Other Gram-negatives’

We think the groups as currently presented make sense because they group species with similar epidemiological patterns, however we appreciate your point that one might initially interpret “Other Gram-negs” as including *Klebsiella* spp. and have therefore inserted the following to clarify:

Lines 156-158: This was also the case for other Gram-negative species (i.e. non *E. coli* and *Klebsiella* spp.) both overall (IRRY: 1.07, 95%CI: 0.93-1.24, p=0.33) and in the neonatal group (IRRY: 0.83, 95%CI: 0.65-1.07, p=0.19).

Line 156 ‘slight’ seems redundant. The analysis suggests a reduction of ~10% per year in children and 20% in neonates.

We agree and have deleted this word.

Would a population-based denominator be more informative for presentation of the relative incidence by age?

We used total pediatric admissions as an offset because we think this might better account for ascertainment than total population (we also do not readily have access to accurate figures for the latter and it is non-trivial to accurately calculate the total catchment area of our hospital).

I found it hard to relate the proportions of resistant isolates in the paragraph starting at line 167 to the paragraph above which gave the breakdown by age. For example, which age group does ‘older children (> 1 month)’ relate to? 123 isolates are reported (24/103 resistant to CRO, and 20 missing), but I can’t work out how this figure was arrived at if there are 66 infants > 1 month, 62 aged 1-4y, 28 aged 5-9y and 47 >10y.

Thankyou for spotting this, the percentage (13%) was correct but the denominator was a typo - 103 should have read 183 (66+62+28+47=203 total patients with 20 missing isolates). We have corrected as follows:

Lines 173-175: “In older children (>1 month), ceftriaxone is the empirical agent used in our setting and in this population 24/183 (13%) isolates were resistant (missing data for 20 isolates).”

Line 224 which compares proportion of E coli/Klebsiella carrying a gene conferring resistance to CRO reports 19/137 relevant E coli, but this is 20/133 in the E coli section above (line 201).

We apologise for the mistake in the denominator and we have carefully checked the numbers in these paragraphs to ensure they are now correct. Part of the issue here was that in line 201 we were comparing either cefotaxime or ceftriaxone resistance whereas in 224 we were comparing only ceftriaxone resistance. The reason for this choice was that many *Klebsiella* spp. have intrinsic *bla_{SHV}* alleles which ResFinder classifies as conferring cefotaxime but not ceftriaxone resistance. The interpretation of phenotypic spectrum for resistance genes is non-trivial and so we thought that excluding these provided a more meaningful comparison. However, we agree it introduces confusion and so have now only included comparisons for alleles that ResFinder reports as conferring ceftriaxone resistance.

Discussion:

Line 275. It would be worth relating this statement to Oxfordshire.

We agree this was not explicit and have amended as follows:

Lines 281-282: “The incidence of these infections has remained stable in Oxfordshire over the past decade in children, in contrast to the increase observed in adults.”

Line 287 this assertion doesn't seem to be supported by the results presented.

We agree and have removed this.

Line 318 'Relative success story' seems an odd turn of phrase for an observational study.

This was intended as a comparison to similar data we have recently published in adults in the same region, however we agree it may be out of context here and have removed.

I found the bars at the top of Figure 3 'representing the carriage of genes producing enzymes with activity against...' very confusing. While Figure 3 presents a great deal of data, I wonder if the presentation of all species on the single tree risks obscuring important information.

We appreciate your perspective however it also may highlight shared gene reservoirs/horizontal gene transfer between species which would be lost if we were to make a tree for each species.

We agree that the coloured bars were confusing and have amended these and clarified the legend which we hope is now clear.

Minor comments:

I would remove 'sepsis' from the running title. Sepsis is really a different clinical phenomenon not specifically relevant to the data presented.

We agree and have removed sepsis from the running title.

Presenting age in days in Table 1 is messy.

We do agree however the breakdown of age is more important in the younger groups where days is more informative. For older children this is less useful but we feel it would be confusing to use e.g. days and years in the same row.

Reviewers' comments:

Reviewer #1 (Remarks to the Author):

the authors have adequately responded to my comments

Reviewer #2 (Remarks to the Author):

Review of revision:

April 2022

General comments and recommendation:

The authors have addressed my comments, and I agree with their responses. The manuscript is an informative and well written piece of work.

I do have two comments which I regret I did not identify in the original manuscript.

Major comment:

The statistical models relating virulence and resistance scores to age are incorrect and ought to be addressed (line 145-). The authors use median quantile regression, but the data are non-negative count data. As illustrated by Figure S3, the linear models used would predict negative virulence and resistance scores (and indeed age) which is clearly impossible. The analysis should be performed using a suitable glm method, such as poisson or negative binomial regression, assumptions tested and diagnostics performed.

Minor comment:

In Table S2 for some reason the rows reporting S/R phenotype for Piptazobactam and Fosfomycin have been inverted compared to the other antimicrobials.

Major comment:

The statistical models relating virulence and resistance scores to age are incorrect and ought to be addressed (line 145-). The authors use median quantile regression, but the data are non-negative count data. As illustrated by Figure S3, the linear models used would predict negative virulence and resistance scores (and indeed age) which is clearly impossible. The analysis should be performed using a suitable glm method, such as poisson or negative binomial regression, assumptions tested and diagnostics performed.

Thankyou for your additional comments which we have carefully considered in consultation with one of the senior authors (Sarah Walker) who is a professor of medical statistics. We did consider the use of a glm model as suggested, however believe there are valid reasons for the choice we have made which we don't think is as clear cut as the reviewer suggests. The intrinsic assumption of the suggested models (e.g. poisson/negative binomial regression) is a log-linear relationship between the outcome (resistance score/virulence gene count) and age – this is how they avoid “negative” age predictions, by essentially forcing this through a log relationship (constant effect per fold change in age which can never reach zero by definition). The negative binomial model also inflates the variance to account for overdispersion. Fundamentally these are just different models with different assumptions and are not more correct than the models we use.

In contrast quantile regression – effectively a regression of a median – is a non-parametric method so doesn't assume any distribution for the outcome. We could, for example, have used a Kruskal-Wallis test to compare across discretised age-group categories (similar to what was done in the French study we cited - 10.1097/INF.0000000000000309), but this would have been underpowered and inefficient. Median quantile regression is essentially just a generalisation of an underpowered Kruskal-Wallis test. Whilst the reviewer is correct to point out that it does not force the 95% CIs to be greater than 0, nor does any regression method that isn't modelling log or logit outcomes. In response to your comments we have added additional lines to Figure S3 to show trend lines and confidence intervals fitted by negative binomial regression (the data is over-dispersed) to illustrate the point we make above.

Minor comment:

In Table S2 for some reason the rows reporting S/R phenotype for Piptazobactam and Fosfomycin have been inverted compared to the other antimicrobials.

Apologies we have fixed this, thankyou for spotting.

Reviewers' comments:

Reviewer #2 (Remarks to the Author):

I re-iterate my earlier statements regarding the overall quality and importance of the manuscript.

That said, I am unconvinced by the response to my criticism of the use of median quantile regression to model resistance and virulence scores. That the model can predict a negative resistance score for the first ~1000 days of the study should indicate to the authors that the approach is wrong. Not only wrong, but in this case probably unhelpful.

As a non-specialist, I acknowledge the credentials of the Professor of medical statistics cited in the rebuttal, but I stand by the criticism and think the data could be better analysed and presented.

Best wishes,

Reviewers' comments:

Reviewer #2 (Remarks to the Author):

I re-iterate my earlier statements regarding the overall quality and importance of the manuscript.

That said, I am unconvinced by the response to my criticism of the use of median quantile regression to model resistance and virulence scores. That the model can predict a negative resistance score for the first ~1000 days of the study should indicate to the authors that the approach is wrong. Not only wrong, but in this case probably unhelpful.

As a non-specialist, I acknowledge the credentials of the Professor of medical statistics cited in the rebuttal, but I stand by the criticism and think the data could be better analysed and presented.

Best wishes,

We thank the reviewers for taking the time to consider our response once again.

The method presented was not intended to predict anything, rather to model trends in the data which we thought were interesting and replicate what has previously been described. When negative binomial regression is used to model the data (to account for the overdispersion observed), age is not significantly associated with resistance or virulence score (however, as previously mentioned this is because the negative binomial model inflates the variance to account for overdispersion). We do not feel comfortable to report that there is no association but appreciate that the reviewers do not agree with the analysis we have performed and have found this unhelpful. Given that we do not see these findings as a key part of what we are trying to communicate overall in the manuscript, we therefore propose removing these sections and have done so in the latest submitted revision.